# Wombs for rent: Exploring the motivations behind Ghanaian Women's decisions to become surrogate mothers

Isaac Mensah Boafo[1], Doris Ayeley Amarteifio[2], Peace Mamle Tetteh[1]*, Rosemond Akpene Hiadzi[1]

1 Department of Sociology, University of Ghana, Accra, Ghana, 2 Ghana Health Service, Accra, Ghana

* ptetteh@ug.edu.gh

## Abstract

This study explores the motivations behind Ghanaian women's decisions to engage in surrogacy, a growing practice within the country's assisted reproductive technology landscape. Using a phenomenological qualitative approach, 21 surrogates from three privately owned agencies in Accra were interviewed to uncover the underlying factors influencing their choices. Thematic analysis, guided by Braun and Clarke's framework, revealed three primary motivators: socio-economic factors, altruism, and religious beliefs. Financial incentives emerged as a significant driver, with participants citing surrogacy as a means to alleviate economic hardship, fund education, or support family needs. However, altruistic motivations were also prevalent, as some women expressed a deep empathy for childless couples and a desire to provide them with a sense of familial completeness. Religious motivations, rooted in Christian doctrines of love and service, further shaped the decisions of a subset of participants, who perceived surrogacy as a moral duty aligned with their faith. The findings underscore the complex interplay between economic vulnerability, cultural norms surrounding motherhood, and ethical considerations in shaping surrogacy decisions. The study situates these motivations within Ghana's socio-economic context, highlighting how financial necessity intertwines with seemingly altruistic and religious frameworks. It concludes that while altruism and spirituality often surface as explicit motivations, economic realities play a critical underlying role, reflecting broader dynamics of commodification in global reproductive markets. This research contributes to a nuanced understanding of surrogacy in Ghana, providing insights into the ethical and socio-economic dimensions of this reproductive practice.

## Background

Surrogacy, a growing form of assisted reproduction, has been subject to ethical, legal, and social scrutiny worldwide, especially in the context of low- and

**Data availability statement:** Data can be accessed at the DOI below: Reserved DOI: 10.17632/dbnknwrvv6.1 We confirm that this DOI is functional and sufficient to be used to access the data. We confirm also that the names of participants and locations in the repository are pseudonyms. The data meets all institutional ethical requirements of anonymity and confidentiality.

**Funding:** The author(s) received no specific funding for this work.

**Competing interests:** The authors have declared that no competing interests exist.

middle-income countries [1–4]. Surrogacy is an advanced assisted reproductive technology that creates pathways to parenthood for various individuals facing reproductive challenges. This includes women with uterine abnormalities, those without a uterus, or those with medical conditions that prevent them from safely carrying a pregnancy. Additionally, it offers a solution for women who have experienced recurrent miscarriages or have undergone unsuccessful fertility treatments. In recent years, surrogacy has also enabled male couples and single individuals to conceive children who are genetically related to them by involving a surrogate who carries the pregnancy. This option represents a significant development in reproductive medicine, allowing individuals and couples, regardless of gender or partnership status, to fulfill their desires for biological parenthood [1].

The duty of procreation to populate the world for the continuity of the human race is not something that societies and individuals take with levity, especially, in Africa [5]. In Ghana, for instance, motherhood is deeply intertwined with cultural identity, and women who struggle with infertility often face considerable social stigma [6,7]. Surrogacy, therefore, becomes not only a medical solution but a social intervention [8], offering hope to childless couples. This is because it affords them the opportunity to have a child (ren) and thus avoid the stigma associated with childlessness.

There are two primary types of surrogacies namely traditional and gestational. In traditional surrogacy, the surrogate mother is also the biological mother of the child, as her own egg is fertilized, often using the sperm of the intended father. In contrast, gestational surrogacy involves implanting a fertilized embryo from the intended parents (or donors) into the surrogate, meaning she has no genetic link to the child [9–11].

While surrogacy provides an option for individuals or couples who may be unable to conceive on their own due to health, fertility, or other reasons, it raises numerous ethical and legal considerations [8,12]. For instance, surrogacy agreements are often complicated by issues surrounding consent, financial compensation, and parental rights, which can vary significantly across legal jurisdictions. The surrogate herself can experience a range of psychological and emotional impacts. Separation from a child she carried for nine months may evoke profound emotional challenges, including grief and loss, regardless of her lack of genetic connection to the child [13]. Additionally, pregnancy itself poses inherent risks, including physical health complications and potential impacts on mental well-being. Despite these challenges, studies have found that many women decide to become surrogates due to various motivations, including altruism, financial incentives, and a desire to help others achieve parenthood [11,14].

However, the motivations of women who choose to become surrogate mothers in Ghana have received less attention from researchers. This study reported herein, sought to explore the complex and multi-faceted factors that motivate women in Ghana to become surrogates.

While many surrogates outside Sub-Saharan Africa emphasize altruistic and religious motivations [15,16], such as the desire to help others or fulfill moral and spiritual duties, these decisions must be analyzed through the lens of commodification. This is because, a surrogate receives payment upon the successful birth of the

child/children she carried in her womb. Commodification thus refers to the process by which something that traditionally holds intrinsic, non-monetary value—such as pregnancy or motherhood—becomes a product to be bought and sold within the market economy – in this case the fertility clinic/hospital/agency. In the context of surrogacy, a woman's womb and reproductive capacity become commodified, exchanged for financial compensation. Thus, even when surrogates frame their decisions in terms of empathy, altruism, or religious obligations, the economic underpinnings of these choices cannot be overlooked [17].

Ghana's socioeconomic landscape is characterized by widespread poverty and limited employment opportunities for many women, particularly those from lower socioeconomic backgrounds [18]. The commodification of surrogacy in such an environment is evident when women, despite expressing non-monetary motivations, operate within an economic system where financial compensation plays a crucial, if unspoken, role. For many surrogates, the financial reward, even when considered secondary, can be the decisive factor that allows them to participate in surrogacy, particularly when they are facing economic hardship.

This study draws on the concept of commodification to argue that while altruistic and religious motivations are present, they are inextricably linked to economic needs. In the context of surrogacy in Ghana, economic survival or advancement often serves as the foundation upon which other motivations are built. By analyzing the narratives of surrogates through the lens of commodification, this research sought to uncover the subtle yet significant ways in which financial need undergirds the decision to become a surrogate. This perspective not only broadens our understanding of surrogacy in Ghana but also highlights the economic realities that shape seemingly altruistic choices within a globalized reproductive market.

## Methods

### Research design and setting

The study reported in this paper is part of a larger study aimed at exploring the lived experiences of surrogate mothers in Ghana. In line with the aim of the research project, the phenomenological qualitative research design was employed. It involved 21 participants drawn from three privately owned surrogate agencies within the Greater Accra region of Ghana. These agencies, which recruit surrogates for intended parents were specifically located in Tema, Osu and Labone. The Greater Accra Region was chosen because it is home to 15 of the 19 in-vitro fertilization (IVF) health facilities in Ghana.

### Sampling

The three agencies were selected purposively through the Fertility Society of Ghana. After submitting the ethical clearance letter of the study and briefing the secretary of the Fertility Society of Ghana, we were given the contact details of five surrogate agencies out of which three agreed to support the study. The three agencies, which for ethical purposes have been named Agency L, Agency O and Agency T offer various services to support the specialist fertility clinics in the management of infertility including gamete donation and surrogacy.

Participants for the study were thus surrogate mothers in Accra who were recruited by the three surrogate agencies from any part of Ghana, and who were either currently pregnant as surrogate mothers (gestational age of 16 weeks and above) or have delivered from surrogate motherhood within two years of providing the surrogate services. We were of the view that these categories of recent surrogate mothers would be able to accurately and comprehensively recall their experiences and motivations. More so, surrogates who were less than 16 weeks pregnant were excluded from the study because it was deemed that they did not have enough experience with antenatal care due to their early gestational age.

Participants who met the eligibility criteria were recruited through three surrogate agencies that served as the primary points of contact. The agency heads first identified eligible surrogates and explained the purpose, scope, and voluntary nature of the study to them. Surrogates who expressed willingness to participate were then given the option of sharing their contact details with the researchers for direct follow-ups. For those who preferred not to disclose their personal

contact information, arrangements were made for the interviews to be conducted using the agency heads' phones. All interviews were thus conducted by telephone, either directly with the participants or via the agency head's device.

In all, N = 21 surrogates were included in the study (Table 1). The sample size was largely determined by data saturation. Interviews lasted between 40 minutes to an hour. Interviews were conducted in Ga, Twi or English. Seventeen interviews were conducted in Twi, three in English and one in the Ga language due to interviewer's and participants' proficiency in these languages. With the permission of participants, all interviews were audio recorded.

With respect to socio-demographic characteristics, N = 10 out of the N = 21 participants were between the ages of N = 26 and N = 30. Regarding ethnic background, N = 10 identified as Akan and N = 7 as Ga-Adangme. Only six participants were married (Table 1). It is also important to note that all participants had at least one child, as this was a criterion used by the recruitment agencies (not shown in Table 1).

## Ethical considerations

The study was approved by the Ethics Committee for the Humanities, University of Ghana (ECH 039/20–21).

Participants were adequately informed about the purpose of the study, the benefits of participation, and their rights as participants. They were assured of the confidentiality of their participation and responses, and informed that they could withdraw from the study at any time without any consequences. Participants were also informed that no information they provided would be directly attributed to them. After this explanation, they were asked a direct question: *"Do you agree to participate in this study under the conditions I have just explained?"* Participants had to respond in the affirmative before

**Table 1. Socio-demographic Characteristics.**

| Item Description | n |
|---|---|
| **Age:** | |
| 20-25 | 3 |
| 26-30 | 10 |
| 31-35 | 5 |
| 36-40 | 3 |
| **Ethnicity:** | |
| Akan | 10 |
| Ga Dangme | 7 |
| Ewe | 3 |
| Frafra | 1 |
| **Marital Status:** | |
| Married | 6 |
| Not married | 15 |
| **Level of Education:** | |
| Junior High School | 7 |
| Senior High School | 12 |
| Tertiary | 2 |
| **Occupation:** | |
| Trading | 8 |
| Hairdressing | 5 |
| Waitress | 2 |
| Cleaner | 1 |
| Unemployed | 3 |
| Student | 2 |

the interview could proceed. Verbal informed consent was thus obtained from each participant, as the researcher(s) did not have the opportunity to meet participants face to face. Participants verbal expressions of agreement to participate in the study were recorded in writing by the interviewer. We did not have other people present to witness the verbal consent given by participants because of the sensitive nature of the subject of surrogacy in Ghana and our bid to provide privacy for our participants. The use of verbal informed consent was approved by the ethics committee as part of the study protocol.

Again, to ensure anonymity of both the facilities where participants were recruited and the participants themselves, these have been given pseudonyms in this paper. Given that the data was obtained through telephone interviews, participants were compensated with fifty cedis (approximately $5 at the time of data collection) to cover the cost of data they expended on each interview call.

## Data analysis

Thematic analysis was employed in analysing the data. The analyses were guided by the steps suggested by Braun and Clarke [19]. The data analysis began with the transcription of the interviews. The transcription of the interviews was completed by author two and one research assistant who had been given some training in the ethics of confidentiality. All interviews were transcribed verbatim. Interviews conducted in the two local languages were transcribed directly into English. Since all the authors are proficient in both languages, we separately compared the transcribed data with the audio recording to ensure that the meanings of participants were not changed. The interview transcripts were then uploaded into the qualitative data analysis software, NVivo (Version 10). The interview transcripts were then read multiple times and where necessary, the audio recordings of the transcripts were played to rectify errors in the transcripts. Through this process of reading and re-reading, the researchers became familiar with all aspects of the data, and ideas for coding were noted. After this familiarization with the data, actual coding of the data began using NVivo. As far as possible, every data item was given equal attention, and as many codes as possible were created. The various codes were placed under appropriate potential themes in the form of parent nodes in NVivo. The development of the themes was theory driven [19] as they were created with specific research questions in mind.

The extracts under each code and theme were read to ensure that they fit into where they were placed. The themes themselves were refined – where necessary, themes were modified or changed to reflect the codes they contain and to ensure minimal overlap between themes.

Some of the codes became themes and others became sub-themes. After this exercise, the data was searched again with the aim of coding any additional data that was missed during the coding process. Codes which were found not to fall under any of the themes of interest were disregarded. The themes that remained captured the data from participant responses and were pertinent to the overall premise of the study itself. The analyses mainly involved searching through the entire data set to identify repeated patterns of meaning. The findings presented in the current paper relate to the motivations for engaging in surrogate motherhood. All extracts presented in the current paper were produced verbatim.

## Findings

Participants in this study had made the rational intentional choice to be surrogates. We sought to investigate both the centripetal and centrifugal factors that led them to make this decision and actually carry it through. As seen in the narratives below, the motivations are complex and hover around three main themes namely: socio-economic, altruism and religion.

## Socio-economic/ financial motivation

The majority of participants in this study—N = 13 out of N = 21—indicated that financial gain was the primary factor driving their decision to participate in surrogacy. Five interviewees explained that they needed the money to upscale their businesses, two needed money to pay their children's school fees, two needed the money to pay their own school fees, one

needed the money to top up what she and the husband has saved so they can pay the rent for their accommodation and three wanted to start trading with the compensation promised for the contract. Their responses shed light on a range of economic challenges that pushed them toward this option. For some women, the financial rewards offered through surrogacy represented a start-up capital for their businesses (N = 3), capital to expand their businesses which had stagnated due to a lack of capital (N = 5). Others were motivated by the pressing need to pay for their children's education (N = 2), as they struggled to meet the costs of school fees. A few women sought financial assistance to continue their own education (N = 2), recognizing that without this support, they would be unable to advance academically or professionally. In one case, a participant saw surrogacy as the only way to contribute to her household's rent, preventing eviction and ensuring the family had a stable living environment. These financial pressures were not merely incidental but fundamental to the participants' decisions, reflecting the socioeconomic vulnerabilities they faced. The women's accounts illustrate how surrogacy offered a potential lifeline in the face of limited employment opportunities, high living costs, and the struggle to meet both personal and family-related financial obligations. The following direct quotes from participants highlight their financial motivations:

Aba, a 27-year-old trader, explained:

For me, it was because of the money that I went in for it. It's not easy carrying a pregnancy for nine months and giving birth for someone, but I needed the money. That's why I did it. My friend told me about surrogacy when I asked her if she knew someone who could lend me money to expand my provision shop, which was struggling. I agreed because I wouldn't have any debt afterward. (Aba, 27-year-old trader)

When asked about her motivation, Delight, a 29-year-old unemployed mother, shared:

At that point, I desperately needed money to take care of my children. I was unemployed and had to pay my children's school fees and start a small business. I had no other choice. (Delight, 29-year-old unemployed)

Emily, a married woman with two children, needed money to supplement her husband's earnings and renew their rent. She explained:

Madam, we were in a difficult situation. The landlord was after us for the rent, and things were really bad financially. When I discussed it with a friend, she told me about surrogacy, and I decided to do it to get the money we needed. (Emily, 28-year-old trader)

Ursula, a 25-year-old trader, expressed her motivation for becoming a surrogate:

The truth is, I wanted money to start my own business and stop working at the waakye joint, where the pay was poor. When a friend told me about surrogacy, I thought it over and decided to go for it. (Ursula, 25-year-old trader)

The financial motivations expressed by these women align with the existing literature on surrogacy, which often identifies economic necessity as a primary driver. Karandikar, Gezinski [20] observed similar motivations in their study of gestational surrogacy in Gujarat, India, where women cited economic desperation and the need for financial stability as the main reasons for their involvement. This parallels the experiences of the women in the current study, who viewed surrogacy as a way to address pressing financial needs, such as educational expenses, rent, and business ventures. The financial compensation offered in surrogacy agreements, therefore, serves as a powerful incentive, especially in contexts where women face limited economic opportunities.

Subedi [4] critiques this transactional nature of surrogacy, noting that it transforms the human body into a commodity within the global reproductive market. In Accra, as in Nepal, the economic vulnerability of women makes

them susceptible to engaging in surrogacy as a form of financial survival. The commodification of women's bodies in this context is starkly evident, as the decision to become a surrogate is directly tied to alleviating financial hardships. The narratives of women like Aba, who needed money to revitalize her struggling provision business, and Delight, who sought funds to pay her children's school fees, illustrate the transactional nature of surrogacy in these settings.

Rozée, Unisa [21] highlight the social paradoxes of commercial surrogacy, particularly in developing countries like India. They argue that while surrogacy can provide significant financial benefits to women, it also perpetuates inequality and exploitation. This duality is present in the experiences of the women in Accra. On one hand, surrogacy offers a solution to their immediate financial needs; on the other, it exposes them to the commodification of their bodies, possible health risks during pregnancy and delivery and thus raising ethical concerns about the exploitation of economically disadvantaged women.

Furthermore, Rozée, Unisa [22] emphasize that surrogacy, particularly in developing countries, often emerges as an option for women with few alternatives. This is consistent with the findings of the current study, where many of the women had no other viable means of securing the necessary funds. For example, Nana, a university student, and Ursula, a trader, both felt they had limited options to improve their financial situations, leading them to accept the offer of surrogacy. This lack of choice reflects broader structural inequalities that shape the decision-making process for women in economically constrained environments.

In economically advanced countries such as the United States, Canada, the United Kingdom, and Israel, research consistently highlights altruism—particularly the desire to help others—as the dominant motivation for women who choose to become surrogate mothers [13,23,24]. For example, a large-scale study in the United States involving 231 participants found that women who become surrogates generally earn above the average income for their state of residence, have higher levels of education, are employed, and possess health insurance. The study further demonstrated that their decision to pursue surrogacy is primarily driven by prosocial and altruistic reasons, rather than poverty or social marginalization [25]. Similar characteristics were also reported in the UK by Horsey, Arian-Schad [26]. Nonetheless, financial considerations cannot be completely dismissed, as some studies suggest that improving one's economic situation remains a relevant factor for certain women [27].

By contrast, in developing countries, financial incentives are most frequently cited as the primary motivation for engaging in surrogacy [3,28,29]. In India, for example, commercial surrogacy became widespread until its prohibition in 2021, after which only altruistic surrogacy was legally permitted [3]. Findings from the current study reflect this broader trend: with the exception of two participants who were students in tertiary institutions, all surrogate mothers had less than tertiary-level education. Moreover, they were either unemployed or employed in occupations considered low-status within the Ghanaian context.

The contrast between these two contexts underscores how socio-economic conditions shape surrogacy motivations, with financial necessity being a dominant centrifugal force that drives women into this space and the cash remunerations, a key centripetal force in developing countries.

## Altruistic motivation

The data further suggests that altruism is one of the motivating factors for becoming a surrogate mother. Some of the participants (N = 6) explicitly stated that their primary motivation for becoming surrogates was driven by altruism and/or empathy. These altruistic motivations demonstrate an acute awareness of their society's expectations regarding fertility and family life. In many instances, these women view surrogacy as a way to contribute positively to the lives of couples who are unable to have children, recognizing that in the Ghanaian culture, infertility can lead to marginalization, particularly for women. As such, some of the surrogates saw the financial reward as a secondary gain, rather than the primary reason for their participation in the surrogacy arrangement.

The following quotes from participants illustrate these points. For example, Gina, a 26-year-old hairdresser, explained her motivation for becoming a surrogate as follows:

My sister approached me about a friend who needed a surrogate. I was touched by their situation, especially when I learned that the woman had experienced five miscarriages. I had also read about surrogacy and had some knowledge of it already. That's why I decided to do it. (Gina, 26-year-old hairdresser)

Gina's response highlights how empathy for the commissioning mother's repeated pregnancy losses, coupled with her existing knowledge of surrogacy, drove her decision.

Another participant, Joyce, a 31-year-old trader, acknowledged that while she was in financial need, her primary motivation was emotional. She shared:

It wasn't really about the money, even though life was tough, and I needed the money to support myself and my children. But seeing someone who can't give birth makes me feel bad. For instance, if I can get pregnant and later want to abort the baby when someone is also looking for a child, I feel it's better to carry the baby and give it to them. It wasn't solely for the money that I agreed to do it. (Joyce, 31-year-old trader)

Joyce's response reflects the internal conflict many surrogates face when they are financially strained but are ultimately moved by compassion for couples struggling with infertility. This brings into the discourse, the notion that motivations may be primary and/or secondary. Thus, while altruism may be the primary reason, personal financial gain remains a significant secondary factor. Conversely, it is possible that financial considerations may be primary, while altruism is secondary for many surrogate mothers.

Botor, a 38-year-old cleaner, similarly expressed empathy as her driving force for becoming a surrogate:

My nurse friend told me about a woman they had been treating who needed a surrogate. The doctors had tried to help her carry her own baby several times, but it failed, maybe four times. I felt sorry for her and decided to help because I have two children of my own. If I can help someone have a child, why not? I can imagine what she must be going through in her marriage because she doesn't have a child. (Botor, 38-year-old cleaner)

When asked if financial motivations played a role in her decision, Botor responded:

As for the money, it was secondary because the amount they gave me wasn't worth the work I did as a surrogate. The main thing that motivated me was sympathy for the woman. Otherwise, I wouldn't have done it. (Botor, 38-year-old cleaner)

The interviews with surrogates in the current study thus offer a profound insight into the altruistic motivations that drive women to become surrogates, reflecting a deep-rooted sense of empathy and fellow feeling. This finding resonates strongly with the broader literature on altruistic motivations in surrogacy, especially in contexts where cultural expectations surrounding motherhood are significant.

Akande [5] discusses how, in many African societies, the role of a mother is central to a woman's identity and social standing. Women who cannot conceive often face stigmatization and marginalization, both within their families and communities. This cultural context thus undergirds the desperation of infertile couples, especially women, making these surrogates want to provide some support to mitigate the imagined and real stigma and ridicule they tend to face. The women's decision to become surrogates, as demonstrated by Gina and Botor, is driven by a desire to alleviate the emotional and social burden faced by childless women in their communities. Their empathy stems from an understanding of the cultural

pressures and the profound impact of infertility on women's lives. Gina, for instance, was moved by the commissioning mother's multiple miscarriages, while Botor felt compelled to help a woman facing the societal shame associated with childlessness. Their altruism is deeply rooted in the cultural imperative to support other women in achieving motherhood, reflecting a sense of solidarity and responsibility.

Gunnarsson Payne, Korolczuk [30] also explore the emotional and relational dimensions of surrogacy, emphasizing how surrogates often report altruism and empathy as their primary motivations. In their critical review, they argue that for many surrogates, the emotional connection with commissioning parents, and a desire to help them overcome infertility, outweighs any financial considerations. This aligns with the experiences of the surrogates in Accra, who view their role as an opportunity to bring joy and fulfilment to couples struggling with infertility. The interviews reveal that surrogates like Joyce and Botor believe their actions will not only help childless couples but also bring blessings to them in the future, reinforcing the altruistic nature of their motivations.

## Religious motivation

Two of the Christian surrogates explicitly cited religious beliefs as their primary motivation for becoming surrogates. Their responses indicated a strong alignment between their Christian faith and their decision to participate in surrogacy. Specifically, they viewed surrogacy as an act of kindness and service to others, in accordance with Christian teachings about helping those in need. This religious motivation was deeply intertwined with the surrogates' belief that their actions would bring them spiritual blessings, which outweighed any financial considerations.

Irene, a 27-year-old waitress, explained her motivation for becoming a surrogate as rooted in her Christian faith:

As a Christian, I believe in doing good; the Bible teaches us to do good. That was what motivated me to be a surrogate mother for them, not because of money, though I was going through financial struggles. But I wouldn't use that means to make money. (Irene, 27-year-old waitress)

Harriet, a 31-year-old trader, also drew a direct connection between her religious beliefs and her role as a surrogate:

The Bible says, love your neighbour as yourself, and I have brought a human being into this world for someone to be happy. It's not because of money, even though I was going through a lot financially when I agreed to be a surrogate. I think God will bless me for helping some couple get a child of their own. (Harriet, 31-year-old trader)

The religious motivations expressed by these surrogates align with broader literature on the role of religion in altruistic behaviour. Bromfield and Rotabi [31] discuss how religious beliefs can strongly influence decisions related to surrogacy, particularly in contexts where individuals see their actions as fulfilling a moral duty. In this case, both Irene and Harriet saw their decision to become surrogates as consistent with Christian teachings about compassion, helping others, and selflessness, all of which can serve as powerful motivators in the absence of material gain.

Gunnarsson Payne, Korolczuk [30] also explore how religious beliefs may play a role in shaping surrogacy relationships, noting that in some cases, surrogates may frame their decision as a form of spiritual or moral duty. This is evident in the responses of the surrogates in Accra, who linked their actions to Christian doctrines of love and service. Irene and Harriet's belief that their surrogacy would result in divine blessings reinforces the idea that for some, surrogacy is not merely a social or economic exchange but also a spiritual one.

The broader context of surrogacy in religious settings is further supported by Akande [5], who notes that in African societies, religious values often shape perceptions of motherhood, fertility, and family life. For the surrogates in this study, their Christian faith provided a moral framework that justified and even encouraged their participation in surrogacy. As Aina points out, motherhood and the ability to help others achieve it can be seen as an extension of religious duty in many African contexts, where fertility is highly valued and tied to social and spiritual fulfilment.

Although participants expressed altruistic and religious motivations for becoming surrogate mothers, we argue that these motivations must also be viewed in light of their social and economic backgrounds. While many participants emphasized altruism and religious beliefs as their primary motivators, such as wanting to help infertile couples or believing their actions aligned with Christian values, it is crucial to recognize the influence of their socioeconomic conditions on their decisions.

For many surrogates, their financial struggles cannot be entirely separated from their motivations, even if they do not explicitly foreground monetary gain. Irene and Harriet, for instance, both mentioned that they were facing financial difficulties at the time of agreeing to become surrogates. Although they downplayed the role of money, describing their motivations as largely altruistic or religious, the fact that they were in financially precarious positions suggests that surrogacy also provided an economic opportunity. Thus, while these women framed their decisions within a religious or moral framework, their economic realities most likely influenced their willingness to participate in surrogacy, even if indirectly.

This phenomenon is well-documented in the literature. Bromfield and Rotabi [31] discuss the role of economic vulnerability in surrogacy decisions, particularly in regions where poverty is widespread. In such contexts, surrogacy may offer women a way to improve their economic situation, even if they do not articulate it as their primary motivation. The financial compensation, although secondary to altruistic or religious goals in the minds of the participants, can still play a significant role in making the decision to become a surrogate. The surrogates in this study may not view themselves as primarily motivated by money, but their financial situations likely made the opportunity to engage in surrogacy more attractive.

Additionally, Gunnarsson Payne et al [30] emphasize the need to consider the socioeconomic context when analyzing surrogacy relationships. They argue that surrogacy cannot be fully understood without recognizing the material conditions that shape individuals' choices. In the case of the surrogates in Accra, the combination of financial strain and religious or altruistic motivations suggests a complex interplay between moral and economic factors. While these women may have viewed surrogacy as a way to help others, their financial difficulties may have also played a role in their decision-making process, even if they did not consciously prioritize it.

Furthermore, Akande [5] highlights that in African societies, economic hardship often intersects with cultural and religious expectations, particularly for women. In contexts where motherhood is highly valued, and women face significant pressure to contribute to their families, surrogacy can be seen as a means to fulfil both moral and economic obligations. This dual responsibility, to help others while also addressing personal financial needs, is a critical factor that must be considered when evaluating the motivations of surrogates. While participants in this study may have expressed altruism and religious devotion, their financial realities likely influenced their ability to act on those motivations. Basically then, the decision to 'rent out their womb' as a surrogate mother has many underlying intertwining factors. Such women tend to be caught in a web of poverty-altruism-religious obligation, leading them to make rational decisions to support infertile others. Thus, in meeting the needs of others, they meet their own needs be they economic or a sense of fulfilment and purpose of having met a religious or moral obligation. It is also worth noting that not all Christian religious groups accept surrogacy. For instance, assisted reproduction is not approved by the Vatican [32] and some other studies have argued that surrogacy is not a Christian practice [2].

## Conclusion

To conclude, although the surrogates in Accra framed their decisions primarily in terms of altruism and religious belief, it is essential to view these motivations within the broader context of their social and economic backgrounds. Economic vulnerability may not have been explicitly cited as a primary motivator, but it likely influenced their decisions. The intersection of financial necessity, altruism, and religious values paints a more comprehensive picture of the complex motivations behind surrogacy in this context, as supported by the literature. Thus, understanding surrogacy decisions requires acknowledging both the moral frameworks that surrogates operate within and the material conditions that shape their

choices. It is important therefore to not lose sight of the complexity of centrifugal and centripetal forces that come to play, either in a parallel and/or fluid pattern in a woman's decision to 'rent her womb out' as a surrogate mother.

## Supporting information

**S1 File. Coding Scheme_Motivations for Surrogacy.**
(DOCX)

**S2 File. Interview Guide-Surrogacy.**
(DOCX)

## Author contributions

**Conceptualization:** PEACE MAMLE TETTEH, Isaac Mensah Boafo, Doris Ayeley Amarteifio.

**Formal analysis:** Doris Ayeley Amarteifio.

**Investigation:** Doris Ayeley Amarteifio.

**Methodology:** Isaac Mensah Boafo.

**Writing – original draft:** Isaac Mensah Boafo, Doris Ayeley Amarteifio.

**Writing – review & editing:** PEACE MAMLE TETTEH, Isaac Mensah Boafo, Rosemond Akpene Hiadzi.

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
