## [Decision Letter · Decision Letter 0]

25 Jun 2025

Dear Dr. TETTEH,

Thank you for submitting your manuscript to PLOS ONE. After careful consideration, we feel that it has merit but does not fully meet PLOS ONE’s publication criteria as it currently stands. Therefore, we invite you to submit a revised version of the manuscript that addresses the points raised during the review process.

Based on the reviewers’ evaluations—one recommending minor revisions and two suggesting major revisions—I am recommending a decision of major revision. The manuscript is thoughtfully developed and demonstrates strong potential for publication. We look forward to receiving your revised submission. Please see the reviewers' comments below. I have also compiled them into a document, which is attached to this message.

We look forward to receiving your revised manuscript.

Kind regards,

Mudasir Mustafa

Academic Editor

PLOS ONE

Journal Requirements:

2. In the ethics statement in the Methods, you have specified that verbal consent was obtained. Please provide additional details regarding how this consent was documented and witnessed, and state whether this was approved by the IRB.

4. In this instance it seems there may be acceptable restrictions in place that prevent the public sharing of your minimal data. However, in line with our goal of ensuring long-term data availability to all interested researchers, PLOS’ Data Policy states that authors cannot be the sole named individuals responsible for ensuring data access (http://journals.plos.org/plosone/s/data-availability#loc-acceptable-data-sharing-methods).

**Additional Editor Comments:**

Minor suggestion: Please proofread the manuscript for grammar and formatting issues before final submission.

Reviewers' comments:

Reviewer's Responses to Questions

**Comments to the Author**

1. Is the manuscript technically sound, and do the data support the conclusions?

Reviewer #1: No

Reviewer #2: Partly

Reviewer #3: Yes

2. Has the statistical analysis been performed appropriately and rigorously?

Reviewer #1: N/A

Reviewer #2: I Don't Know

Reviewer #3: N/A

3. Have the authors made all data underlying the findings in their manuscript fully available?

Reviewer #1: Yes

Reviewer #2: No

Reviewer #3: No

4. Is the manuscript presented in an intelligible fashion and written in standard English?

Reviewer #1: Yes

Reviewer #2: Yes

Reviewer #3: Yes

Reviewer #1: An interesting paper. Many comments are points for clarification and issues to reflect upon.

Table 1 – It would be useful to provide the characteristics of women who undertake surrogacy.

-bold “level of education”

In the results – did culture feature as a theme? I see a bit integrated in the section on altruism but perhaps it deserves to be a separate theme.

Page 11, the content on gay fathers – are these gay couples who are biologically male? So do the females volunteer for surrogacy out of need to support the gay fathers? What kind of women offer to become surrogate mothers in developed countries? It would help to make a comparison.

Of course there are variations between developed and developing world settings. This is why it is important to first describe the characteristics of the respondents. Did the authors conduct interviews or review literature on the general characteristics women that participate in surrogacy? Such would further strengthen the conclusions of the manuscript.

P16, the discussion should include contrary ideas for instance in Christianity where surrogacy is considered unnatural.

Isn’t is possible for poor persons to have altruistic motives? One of them even mentioned that the funds provided did not measure to the service provided. Concerning the overriding motivation being economic – a question that should have been asked is “would they have volunteered without the economic incentive”? This would strengthen the discussion on motives for participation.

Ethics – were participants allowed not to answer questions that they did not wish to answer?

Data – is plural.

Table 1 – please bold the “level of education”

Reviewer #2: Please see my attached document for comments regarding this submission. I believe this is an interesting paper, largely very well written. However, the associated data is largely non-existent. This must be provided for consideration.

Reviewer #3: The study is interesting, providing information about a specific population that was not previously known, but it does not bring anything new or different from what was expected to be found.

The sample size is small, although it is a qualitative study, so it does not allow generalizations to be made to the entire population.

The study was approved by an ethics committee, which accepted informed consent in oral form. Although it was a survey conducted by telephone, it could have been conducted in two stages, with a prior period to explain the study and request the submission of a written informed consent form.

The authors do not provide an interview script, but they should. There are very relevant aspects that are not mentioned, so we are left wondering whether they were asked, such as whether the woman already has children, and how her family reacted to the surrogacy process.

Regarding the motivations for this practice, all participants cite economic reasons as having weight in their decision. It would be important to question whether this decision would be maintained if there were no payment for it.

There are very relevant ethical issues that justify the non-acceptance of surrogacy in many countries. In others, it is permitted, but in an altruistic manner, that is, without payment for this practice. In still others, such as India (one of the few countries cited by the authors), it is no longer permitted for non-resident foreigners. It would be important to delve deeper into these issues, and there is literature available on the subject.

.

Reviewer #1: No

Reviewer #2: No

Reviewer #3: No

---

## [Author Response · Author response to Decision Letter 1]

28 Oct 2025

Comments Reviewer #1 Responses

An interesting paper. Many comments are points for clarification and issues to reflect upon.

Table 1 – It would be useful to provide the characteristics of women who undertake surrogacy.

-bold “level of education”

This has been done. See Table 1

In the results – did culture feature as a theme? I see a bit integrated in the section on altruism but perhaps it deserves to be a separate theme.

Culture did not come up as a theme per se. Indeed, in the Ghanaian society, although childbearing is valued, surrogacy is no largely acceptable and surrogates and commissioning parents risk stigmatisation should they be known. However, the culture generally encourages altruism.

Page 11, the content on gay fathers – are these gay couples who are biologically male? So do the females volunteer for surrogacy out of need to support the gay fathers? What kind of women offer to become surrogate mothers in developed countries? It would help to make a comparison.

Yes, the gay fathers in the study cited were biologically males.

The comparison between developed and underdeveloped countries have also been incorporated in that section. It reads:

In economically advanced countries such as the United States, Canada, the United Kingdom, and Israel, research consistently highlights altruism—particularly the desire to help others—as the dominant motivation for women who choose to become surrogate mothers (Fantus & Newman, 2019; Hohman & Hagan, 2001; Imrie & Jadva, 2014). For example, a large-scale study in the United States involving 231 participants found that women who become surrogates generally earn above the average income for their state of residence, have higher levels of education, are employed, and possess health insurance. The study further demonstrated that their decision to pursue surrogacy is primarily driven by prosocial and altruistic reasons, rather than poverty or social marginalization (Martínez-López & Munuera-Gómez, 2024). Similar characteristics were also reported in the UK by Horsey, Arian-Schad, Macklon, and Ahuja (2022). Nonetheless, financial considerations cannot be completely dismissed, as some studies suggest that improving one’s economic situation remains a relevant factor for certain women (Khvorostianov, 2023).

By contrast, in developing countries, financial incentives are most frequently cited as the primary motivation for engaging in surrogacy (Pashmi, Ahmadi, & Tabatabaie, 2009; Sharma, 2023; Taebi, Alavi, & Ahmadi, 2020). In India, for example, commercial surrogacy became widespread until its prohibition in 2021, after which only altruistic surrogacy was legally permitted (Sharma, 2023). Findings from the current study reflect this broader trend: with the exception of two participants who were students in tertiary institutions, all surrogate mothers had less than tertiary-level education. Moreover, they were either unemployed or employed in occupations considered low-status within the Ghanaian context.

Of course, there are variations between developed and developing world settings. This is why it is important to first describe the characteristics of the respondents. Did the authors conduct interviews or review literature on the general characteristics of women who participate in surrogacy? Such would further strengthen the conclusions of the manuscript.

The characteristics of participants have been described in Table 1. It shows clearly, that almost all of them are from low socio-economic backgrounds. This is also reiterated in their motivations for becoming surrogates.

P16, the discussion should include contrary ideas for instance, in Christianity, where surrogacy is considered unnatural.

This has been done. See the last sentence under Religious Motivation. It reads:

It is also worth noting that not all Christian religious groups accept surrogacy. For instance, assisted reproduction is not approved by the Vatican (Schenker, 2005)and some other studies have argued that surrogacy is not a Christian practice (Saluun & Bunde, 2024).

Isn’t is possible for poor persons to have altruistic motives? One of them even mentioned that the funds provided did not measure to the service provided. Concerning the overriding motivation being economic – a question that should have been asked is “would they have volunteered without the economic incentive”? This would strengthen the discussion on motives for participation.

This is well noted. And we agree that some poor persons could do this out of altruism. This question was not explicitly asked but we can deduce from some responses. For instance, the response of Gina suggests it was possible for her to volunteer even without economic incentive. In contrast, Joyce narrative shows otherwise.

We have also revised part of the discussion:

“Thus, while altruism may be the primary reason, personal financial gain remains a significant secondary factor. Conversely, it is possible that financial considerations may be primary, while altruism is secondary for many surrogate mothers.”

Ethics – were participants allowed not to answer questions that they did not wish to answer?

Indeed, participants were free to decide whether or not to answer a question and to even withdraw from the study entirely. This was clearly explained to all participants.

Data – is plural.

Noted. We have corrected this. All sentences which were written as ‘data was’ has been corrected to ‘data were’.

Table 1 – please bold the “level of education”.

This has been done

Reviewer #2

I found the abstract to be captivating. It was very well written, and it captured the various reasons for surrogacy in the unique cultural environment of Ghana very well.

Thank you

I would suggest citing the very first sentence citing the enhanced scrutiny of surrogacy in low- and middle-income countries. There are other sentences on the final paragraph of page 3 that I would suggest providing a cita)on for, as well. We have cited these sentences as suggested.

“Surrogacy, a growing form of assisted reproduction, has been subject to ethical, legal, and social scrutiny worldwide, especially in the context of low- and middle-income countries (Olaye-Felix, Allen, & Metcalfe, 2023; Saluun & Bunde, 2024; Sharma, 2023; Subedi, 2015).”

I suggest un-bolding of the words ‘traditional’ and ‘gestational’ on page 3. This has been done.

I suggest rewording the concluding sentence on page 5 to read “We were of the view that these categories of recent surrogate mothers would be able to accurately and comprehensively recall their experiences and motivations.” This has been done. See the concluding sentence on page 4 under the heading Sampling.

I would also reword the following sentence to read “More so, surrogates who were less than 16 weeks pregnant were excluded from the study because it was deemed that they did not have enough experience with antenatal care due to their early gestational age.” This has been done.

In discussing how participants were contacted, the authors need to provide explanation how specifically these subjects were contacted through the agencies. Was it by telephone? Email? Social media? (this does happen in studies). The sentence structure in this paragraph overall does not seem as robustly written as the Abstract/Background, perhaps due to a language barrier?

The selection procedures have been made more succinct. It now reads:

Participants who met the eligibility criteria were recruited through three surrogate agencies that served as the primary points of contact. The agency heads first identified eligible surrogates and explained the purpose, scope, and voluntary nature of the study to them. Surrogates who expressed willingness to participate were then given the option of sharing their contact details with the researchers for direct follow-up. For those who preferred not to disclose their personal contact information, arrangements were made for the interviews to be conducted using the agency heads’ phones. All interviews were thus conducted by telephone, either directly with the participants or via the agency head’s device.

Remove the word “See” before (“Table 1”) This has been done

In the ethical considerations section, I would edit the paragraph to read “participants were appropriately counselled on the merits of their participation in this study, the confidentiality of their participation and responses was assured, and it was communicated that they could withdraw from the study at any point without consequence.” The phrase “made to understand” sounds somewhat forceful. Following this, remove the capital “P” from the word “pseudonyms”.

This has been revised as per the suggestion.

I do not believe it is necessary to note that the interviews were conducted by the “second author” or “third author” (both on page 6)…. I would instead simply replace this with “interviewer”.

We have deleted this from the manuscript and revised as per the suggestion.

In data analysis, replace “over and over” with “multiple times”. I would also replace those ideas that coding was “noted”, instead of “jotted down”.

Suggestions have been effected.

Replace “there is not much overlap” with “minimal”. Replace “put aside” with “disregarded”. Replace “the themes that remained are those that tell a story about the data in relation to the research question” with “the themes that remained captured the data from participant responses, and were pertinent to the overall premise of the study itself.” Thanks for the suggestions. They have been effected.

It is troubling that the authors noted that they used Nvivo for data analysis and coding of data, but this data is not included in any way, with no thematic maps included. This information must be provided.

We did not include a coding map because the development of the themes was quite straight forward. We did not have to create parent and child nodes. We are therefore of the view that presenting a table of coding or coding map will simply be a repetition of what has been stated in the methods and the findings

For the socio-economic/financial motivation paragraph, it would be helpful to provide quantification data regarding the responses. For example, it notes that for some women, financial rewards for business purposes was a motivating factor; how many subjects noted this ? How many sought financial assistance for their own education? This data should be presented in a table, and potentially specified in the text. This has been indicated as suggested:

“Their responses shed light on a range of economic challenges that pushed them toward this option. For some women, the financial rewards offered through surrogacy represented a start-up capital for their businesses (3), capital to expand their businesses which had stagnated due to a lack of capital (5). Others were motivated by the pressing need to pay for their children’s education (2), as they struggled to meet the costs of school fees. A few women sought financial assistance to continue their own education (2), recognizing that without this support, they would be unable to advance academically or professionally. In one case, a participant saw surrogacy as the only way to contribute to her household’s rent, preventing eviction and ensuring the family had a stable living environment.”

The discussion which follows is excellent. Thank you.

I would replace the word “gay” with “homosexual” in the last paragraph on page 11. That section has been revised and the word ‘gay’ no longer appears.

I would reword the sentences “Thus, whilst altruism may be the primary reason, the potential to make some money could also not be taken out of the equation. In fact, the reverse could be true where financial considerations may be primary and altruism, secondary for many surrogate mothers.” With Thus, while altruism may be the primary reason, personal financial gain remains a significant secondary factor. Conversely, it is possible that financial considerations may be primary, while altruism is secondary for many surrogate mothers.” Thanks for the suggestion this has been reworded as per your suggestion. See page 12.

Overall, this is an interesting study, much of which is very well written, and I believe it is worthy of publication. However, additional data on patient responses needs to be provided. Thank you.

The study did not involve patients, we are therefore a bit confused about the suggestion that additional data on patient responses need to be provided.

Reviewer #3

The study is interesting, providing information about a specific population that was not previously known, but it does not bring anything new or different from what was expected to be found.

Thank you. However, we do not agree entirely that the study brings nothing new. As you rightly pointed out, it provides information on previously on a practice that has been understudied in Ghana. It also highlights how the socio-economic context within which surrogacy happens influence motivations.

The sample size is small, although it is a qualitative study, the small sample size does not allow generalizations to the entire population. You are absolutely right. The aim of this study is not to make generalizations.

The study was approved by an ethics committee, which accepted informed consent in oral form. Although it was a survey conducted by telephone, it could have been conducted in two stages, with a prior period to explain the study and request the submission of a written informed consent form.

This is well noted. However, as stated in the paper, the study was explained to them prior to the interview albeit not on different occasions as suggested. Moreover, if the reviewer is a bit more conversant with the Ghanaian terrain, he will appreciate that with such a sensitive topic like surrogacy, many participants will not be willing to sign a consent form besides the fact that it may be considered an inconvenience for them to sign and return these forms to the researchers.

The authors do not provide an interview script, but they should. There are very relevant aspects that are not mentioned, so we are left wondering whether they were asked, such as whether the woman already has children, and how her family reacted to the surrogacy process. We do agree with the reviewer that they may have questions on other aspects of the study. We have not provided interview transcripts at this time because the analysis and write-ups are still on-going. For instance, we are preparing manuscripts on the recruitment and discharge processes, the Experiences and Challenges. In the paper focused on the challenges, we, for instance, discusses the issue of stigmatization and how surrogate mothers manage information to remain discreditable. Once these are done, we shall deposit the transcripts in a repository. In the meantime, they can be made available upon reasonable request to the corresponding author.

We have, however, added information as to whether the participants had children. See the last paragraph of Sampling. It reads:

“With respect to socio-demographic characteristics, 10 out of the 21 participants were between the ages of 26 and 30. Regarding ethnic background, 10 identified as Akan and 7 as Ga-Adangme. Only six participants were married (Table 1). It is also important to note that all participants had at least one child, as this was a criterion used by the recruitment agencies (not shown in Table 1).”

Regarding the motivations for this practice, all participants cite economic reasons as having weight in their decision. It would be important to question whether this decision would be maintained if there were no payment for it. For most participants, but for the financial payment, they wouldn’t have opted to be surrogate mothers. See the section on Financial Motivation.

There are very relevant ethical issues that justify the non-acceptance of surrogacy in many countries. In others, it is permitted, but in an altruistic manner, that is, without payment for this practice. In still others, such as India (one of the few countries cited by the authors), it is no longer permitted for non-resident foreigners. It would be important to delve deeper into these issues, and there is literature available on the subjec

---

## [Decision Letter · Decision Letter 1]

26 Nov 2025

Dear Dr. TETTEH,

Remove two dots: See fourth line under heading Ethical Considerations, see “without consequence..”Table on one page: move the paragraph above Table 1 to under Table 1, so table could fit on a page.Space after citation: second last line under religious motivation, see “(Schenker, 2005)and”Fix paragraph spacing.For each quotation, remove period (full stop) after parenthesis to be constant with formatting. E.g. (Emily, 28-year-old trader).Add supplement material as suggested by reviewer, like interview guide. Also coding scheme or word cloud from Nvivo.Please proof read one time more for grammar and sentence structure.

We look forward to receiving your revised manuscript.

Kind regards,

Mudasir Mustafa

Academic Editor

PLOS ONE

Journal Requirements:

Reviewers' comments:

Reviewer's Responses to Questions

**Comments to the Author**

Reviewer #1: All comments have been addressed

Reviewer #2: (No Response)

2. Is the manuscript technically sound, and do the data support the conclusions?

Reviewer #1: Yes

Reviewer #2: Yes

3. Has the statistical analysis been performed appropriately and rigorously?

Reviewer #1: N/A

Reviewer #2: Yes

4. Have the authors made all data underlying the findings in their manuscript fully available?

Reviewer #1: (No Response)

Reviewer #2: No

5. Is the manuscript presented in an intelligible fashion and written in standard English?

Reviewer #1: Yes

Reviewer #2: Yes

Reviewer #1: The authors have adequately addressed the comments. Minor issues: In the section on ethical issues, the authors could consider ethical limitations posed by using the agency's telephone. The respondent may not feel free to discuss issues that may not favor the agency unless details under which the environment and conditions under which the interviews were conducted are provided.

Reviewer #2: This is a solid revision of the original manuscript. I believe it is close to what is needed for acceptance.

.

Reviewer #1: No

Reviewer #2: No

---

## [Author Response · Author response to Decision Letter 2]

20 Feb 2026

All edits and proofreading have been done. The Interview Guide and Coding scheme have been added as supplementary documents. Questions related to verbal consent all addressed in response to reviewers' form. have all been numbered

---

## [Editor Report · Decision Letter 2]

16 Mar 2026

Wombs for Rent: Exploring the Motivations behind Ghanaian Women's Decisions to Become Surrogate Mothers

PONE-D-24-59640R2

Dear Dr. TETTEH,

We’re pleased to inform you that your manuscript has been judged scientifically suitable for publication and will be formally accepted for publication once it meets all outstanding technical requirements.

Kind regards,

Mudasir Mustafa

Academic Editor

PLOS One

Additional Editor Comments (optional):

Check references please: A few references are still in different format, such as on page 19, check this reference *(Sharma, 2023) and see page 25 as well.**As references are numbered, so to avoid confusion of frequency of participants, you may use you may use N. For instance, under heading Altruistic Motivation, “Some of the participants (6) explicitly”, can be changed to “Some of the participants (N=6) explicitly”, check for rest of finding section*.
*Typesetting or formatting: Some paragraphs are single spaced, see page 16. Table could be shrunk to one page by removing paragraph spacing.*
---

## [Editor Report · Acceptance letter]

PONE-D-24-59640R2

PLOS One

Dear Dr. TETTEH,

I'm pleased to inform you that your manuscript has been deemed suitable for publication in PLOS One. Congratulations! Your manuscript is now being handed over to our production team.

Kind regards,

on behalf of

Dr. Mudasir Mustafa

Academic Editor

PLOS One